# Mechanisms That Protect Mammalian Sperm from the Spontaneous Acrosome Reaction

**DOI:** 10.3390/ijms242317005

**Published:** 2023-11-30

**Authors:** Haim Breitbart, Elina Grinshtein

**Affiliations:** The Faculty of Life Sciences, Bar-Ilan University, Ramat Gan 5290002, Israel

**Keywords:** spermatozoa, capacitation, acrosome reaction, spontaneous acrosome reaction, actin polymerization, signaling

## Abstract

To acquire the capacity to fertilize the oocyte, mammalian spermatozoa must undergo a series of biochemical reactions in the female reproductive tract, which are collectively called capacitation. The capacitated spermatozoa subsequently interact with the oocyte zona-pellucida and undergo the acrosome reaction, which enables the penetration of the oocyte and subsequent fertilization. However, the spontaneous acrosome reaction (sAR) can occur prematurely in the sperm before reaching the oocyte cumulus oophorus, thereby jeopardizing fertilization. One of the main processes in capacitation involves actin polymerization, and the resulting F-actin is subsequently dispersed prior to the acrosome reaction. Several biochemical reactions that occur during sperm capacitation, including actin polymerization, protect sperm from sAR. In the present review, we describe the protective mechanisms that regulate sperm capacitation and prevent sAR.

## 1. Introduction

Prior to penetrating the oocyte, mammalian spermatozoa should undergo a highly regulated process called the acrosome reaction (AR). The physiological AR is a precise regulated Ca^2+^-dependent exocytotic process induced by the sperm–oocyte contact, causing a rapid increase in intracellular Ca^2+^ concentrations, thereby initiating the AR [1,2]. It is generally accepted that the physiological AR occurs as a result of the interaction of intact sperm with the oocyte zona-pellucida (ZP). Florman and Storey suggested that the ZP is the site of the AR in mice [3], though it was also suggested that mouse sperm begin to undergo the AR in the upper isthmus of the oviduct [4]. During IVF in mice, acrosome-intact sperm remain attached to the ZP for a longer time than reacted sperm, thereby facilitating fertilization [5,6]. However, it has been suggested that mouse sperm that undergo the AR before contact with the oocyte ZP can still fertilize the oocyte [7]. Pre-treatment of bovine sperm with ZP-glycoproteins causes an increase in the AR and significantly inhibits the subsequent penetration of these sperm into the oocyte, suggesting that the AR occurs after the initial interaction between the sperm and the oocyte, at least in cows [8]. ZP isolated from various species are able to induce the AR in mice, hamsters, guinea pigs, rabbits, cows, monkeys and humans [9].

Before initial contact with the oocyte and in order to undergo the AR, mammalian sperm must first undergo several biochemical processes in the female reproductive tract, which are collectively called capacitation (rev. in [10]). Our group demonstrated that actin polymerization occurs during sperm capacitation and that the F-actin is then dispersed prior to the AR [11]. Inhibition of F-actin formation during sperm capacitation results in the spontaneous acrosome reaction (sAR) [12]. The sAR is a premature form of the AR that does not lead to productive fertilization. It is defined as an AR that occurs in sperm incubated under capacitation conditions but without any AR-inducer, while the physiological AR is defined as an AR that occurs in capacitated sperm after induction by ZP or by other known inducers such as Ca^2+^-ionophores or progesterone. Morphologically, the sAR appears similar to the induced AR. However, sperm samples with a high proportion of cells that have undergone the sAR result in poor success in human IVF [13]. In varicocele patients, the autoimmune antisperm reaction is accompanied by the presence of the sAR and a lack of induced reactions and an increase in intracellular reactive oxygen species (ROS) concentration and DNA fragmentation [14]. Sperm in obese men show a low fertility rate and elevated sAR levels, which are associated with altered circulating levels of estradiol (E2) and sperm cholesterol content [15]. A similar increase in the sAR was seen in spermatozoa from mice fed a high-fat diet [16]. These results suggest that a decrease in E2 and fatty acid levels may influence spermatogenesis [17] and may affect some steps of acrosome biogenesis that will have consequences for fertilization. The molecule 2-arachidonoylglycerol (2AG) affects the in vitro functionality of human sperm by reducing motility, inhibiting capacitation and triggering the sAR [18]. It was shown in human sperm that 2AG inhibits the Ca^2+^-channel CatSper and accumulates in the cell when the progesterone-dependent lipid hydrolase ABHD2 is blocked [19].

The degree of the sAR in human sperm may have clinical importance in predicting the results of IVF, as it is negatively correlated with the achievement via IVF of high-quality embryos and pregnancy rate [20]. Loading sterols into chicken spermatozoa before cryopreservation enhances their quality by inhibiting early apoptotic changes and the sAR [21]. However, sperm from polyzoospermic men demonstrate a low sAR rate as well as low levels of Ca^2+^-ionophore (A23187)-induced AR [22,23,24]. Nevertheless, in boar sperm, the percentage of the sAR was not significantly different in fertile (4.5%) versus subfertile boars (4.75%) [25]. Thus, there are differences among various species regarding the correlation between the sAR and fertilization rate.

In mice, the sAR renders spermatozoa fertilization incompetent [3]. Moreover, an intact acrosome is required for the chemotaxis of mouse spermatozoa towards the oocyte [26], indicating that spermatozoa that undergo the sAR before reaching the oocyte cumulus oophorus are unlikely to respond to the oocyte chemotactic signals. Suarez showed that 98% of rabbit sperm collected from the oviduct ampulla at the beginning of fertilization were acrosome-intact [27], suggesting that acrosome-reacted sperm are unable to penetrate the oviduct. Thus, to achieve fertilization, the sperm must prevent the AR from occurring before contact with the oocyte. The goal of this review is to describe the known mechanisms that protect spermatozoa from the sAR.

## 2. Role of Actin Polymerization

The conversion of G-actin to F-actin is a necessary process to achieve sperm capacitation [11,28]. This process of actin polymerization is mediated by phospholipase D (PLD) and the kinases protein kinase A (PKA) and tyrosine kinase [28], two key kinases involved in the capacitation process [29,30] (see Figure 1). At least three tyrosine kinases are involved in sperm capacitation: Src, Pyk2 and Fer (see Figure 1). We showed in bovine sperm that the PKA and protein kinase C (PKC)-dependent signal transduction pathways can potentially lead to PLD activation; however, under physiological capacitating conditions, actin polymerization depends primarily on PKA activation [28]. The activation of PKA during capacitation causes the inactivation of phospholipase C (PLC), preventing PKC activation [28] (see Figure 1). The role of PLC is to hydrolyze phosphatidyl-inositol-4,5-bisphosphate (PIP_2_) to diacylglycerol, which activates PKC, and inositol-triphosphate (IP_3_), which in turn activates the Ca^2+^ channel in the outer acrosomal membrane, causing Ca^2+^ release from the acrosome, further activating the Ca^2+^-activated PKC.

PKA activation promotes F-actin formation and sperm capacitation, whereas the premature activation of PKC during capacitation jeopardizes this process. Indeed, PKA mediates PLD activation and the inhibition of PKA, resulting in an increase in the sAR and a decrease in F-actin levels, and these two activities can be reversed by adding phosphatidic acid vesicles, the product of PLD activity [31]. The activation of PKC in mouse sperm showed no effect on the sAR [32]. However, the addition of bicarbonate to equine or hamster sperm, which activates the soluble adenylyl cyclase to produce cAMP, leading to PKA activation, causes an increase in the sAR [33,34] in contrast to the findings in bovine sperm described above. Hyperactivated (HA) human spermatozoa generally show high levels of capacitation and display greater sAR levels than non-HA spermatozoa during incubation in synthetic culture media [35]. Moreover, aryl hydrocarbon receptor-KO spermatozoa were frequently capacitated and showed reduced sAR levels and very limited actin polymerization during capacitation [36]. These results in mouse sperm contradict the data in bovine sperm, in which the inhibition of PKA inhibits actin polymerization and capacitation resulted in an increase in the sAR [31]. It is therefore possible that low F-actin levels prevent capacitation in bovine but not in mouse sperm.

In human sperm, the hydrolysis of PIP_2_ by PLC prior to the AR causes the release of the bound actin-severing protein gelsolin, which disperses F-actin, allowing the AR to occur [37]. During capacitation, gelsolin is bound to PIP_2_ and undergoes tyrosine phosphorylation by sarcoma-protein kinase (Src), two processes which keep gelsolin inactive during capacitation, allowing the formation of F-actin [37] (see Figure 1), which protects the sperm from the sAR.

Phosphatidyl-inositol-3-kinase (PI3K) is phosphorylated on tyrosine-467 during bovine sperm capacitation and mediates F-actin formation only when PKA is highly activated (by the addition of exogenous cAMP to the sperm) [38]. PKA activates the tyrosine kinase Src, which inhibits protein-phosphatase 1 (PP1), leading to Ca^2+^/calmodulin-dependent protein kinase II (CaMKII) activation. This in turn mediates PI3K-tyrosine phosphorylation by activating the tyrosine-kinase Pyk2 [39] (see Figure 1). We showed that PKA phosphorylates glycogen-synthase kinase-3 (GSK3), causing its inactivation, leading to low sAR levels. This suggests that the maintenance of acrosome reaction timing is mediated by PKA via the regulation of GSK-3 beta activity [40]. In a recent study, we also showed in human sperm that the inhibition of PP1 by Src is mediated by the tyrosine kinase FER, which is activated by PKA/Src activities [41]. Src-family-kinase (SFK) phosphorylation in bird sperm inhibits the sAR, but interestingly, in birds sperm, SFK is not localized downstream to PKA and is primarily regulated by calcium-dependent tyrosine phosphatase activity [42].

CaMKII induces F-actin formation either by inducing actin polymerization or by stabilizing actin fibers [43]. It was shown that active CaMKII prevents the sAR in mouse sperm by interacting with the multi-PDZ-domain protein1, MUPP1 [44]. The inhibition of CaMKII in bovine sperm results in the sAR, and this effect is reversed by the activation of PLD by spermine [12]. Spermine activates phosphatidyl-inositol-4-kinase (PI4K), resulting in an increase the level of PIP_2_, which is a cofactor for PLD activation. Furthermore, the decrease in F-actin and the increase in the sAR by the inactivation of the PLD pathway can be reversed by CaMKII activation using H_2_O_2_ or PP1 inhibition [12], and the recovery by PP1 inhibition is mediated by PI3K [31]. In order to fully activate actin polymerization and prevent the sAR, both forms of CaMKII, p-CaMKII and oxidized CaMKII should be activated [12]. These results indicate that two distinct pathways, involving PLD or CaMKII, lead to F-actin formation during capacitation and protect the sperm from the sAR.

We found that Ezrin activity during sperm capacitation mediates actin polymerization and prevents the occurrence of the sAR in bovine sperm [45]. Ezrin, Radixine and Merlin are closely related proteins called ERM proteins, which form cross links between the plasma membrane and actin and thereby mediate actin polymerization in cells. Ezrin is highly phosphorylated/activated during the first hour of the capacitation process, and subsequently, its phosphorylation rate is significantly decreased. Ezrin phosphorylation depends on protein PKA and CaMKII activities and to some extent on PI4K activity. The inhibition of these three kinases stimulates the sAR, in which the effect of PI4K inhibition, but not the inhibition of PKA or CaMKII, can be reversed by increasing p-Ezrin using a phosphatase inhibitor [45].

## 3. Ca^2+^ Transport Mechanisms Regulates the AR

Extracellular Ca^2+^ is required to trigger the sAR in human spermatozoa [46]. A relatively low extracellular Ca^2+^ concentration (~30 µM) is required for sperm capacitation’ however, Ca^2+^ is not required for protein tyrosine phosphorylation, and high [Ca^2+^]_i_ (0.15 mM) decreases the protein tyrosine phosphorylation levels [47]. Relatively high [Ca^2+^]_i_ induces the sAR, and therefore the sperm must precisely regulate its [Ca^2+^]_i_. Calcium channels, including the sperm-specific cation channel CatSper, induce the sAR in human and bovine sperm [41,48]. The inhibition of the Ca^2+^-ATPase of the outer acrosomal membrane using thapsigargin results in Ca^2+^ release from the acrosome to the cytosol via IP_3_R, leading to Ca^2+^ influx into the cell via the Ca^2+^-dependent Ca^2+^ channel (CDCC) of the plasma membrane [49], which triggers the sAR [50]. Thus, the Ca^2+^-ATPase of the outer-acrosomal membrane protects sperm cells from the sAR by maintaining a relatively high [Ca^2+^] inside the acrosome and preventing CDCC activity.

NMDA-type glutamate receptor mediates the sAR in newt sperm by increasing Ca^2+^ transport into the cells [51]. Furthermore, in human sperm, the inhibition of the sperm-specific potassium channel, KSPER, decreased the level of sAR [52]. In mouse sperm, KSPER and CatSper together account for all cation currents activated by voltage and alkalization [53] and are thought to act in concert to mediate the changes in membrane cation conductance and Ca^2+^ influx that occur during the onset of capacitation [54].

The decapacitation mechanism of the seminal-vesicle-auto-antigen might target membrane sphingomyelin and regulate the plasma membrane Ca^2+^-ATPase activity to reduce the intracellular Ca^2+^ concentration, thereby reducing the cAMP level and preventing the sAR [55].

Other mechanisms involving Ca^2+^ may also inhibit the sAR. It was suggested that pH-dependent Ca^2+^ oscillations prevent premature sAR in human sperm [56,57]. In addition, it was shown that ~30% of human sperm display spontaneous Ca^2+^ oscillations correlated with the absence of the AR, suggesting another mechanism reducing the occurrence of the sAR. It was also suggested that protein–protein interactions between the Ca^2+^-sensor protein synaptotagmin [58] and SNARE-associated complexin [59,60,61] maintain the membrane fusion machinery at an intermediate pre-fusion stage [62], thereby preventing the sAR [63].

In analogy to Ca^2+^ transport, H^+^ transport mechanisms may also affect the sAR. Alkalization of the intra-acrosomal space was shown to cause an increase in the sAR [64]. These authors suggested that the vacuolar-type H1 ATPase (V-ATPase), the Na^+^/H^+^ exchanger (NHE) and the Cl^−^/HCO_3_^−^ exchanger maintain the acidic pH in the acrosome and prevent inner-acrosomal alkalization and the sAR [64]. Conversely, cytosol alkalization leads to CatSper activation [65] and to elevated sAR levels (our unpublished data). In conclusion, Ca^2+^ transport mechanisms mediate the sAR.

## 4. Role of Reactive Oxygen Species (ROS) and Mitochondrial Activity in the sAR

Oxidative stress is currently considered to be a main cause of male infertility (rev. by [66]). Although presence of a basal level of ROS is essential for the onset of sperm-activating processes such as capacitation [67], its increased levels disturb sperm functions, thereby leading to male infertility by mechanisms such as lipid peroxidation and DNA damage [68]. The levels of ROS are therefore precisely regulated in sperm, mainly by superoxide dismutase (SOD), which coverts superoxide anions to H_2_O_2_ [69], and by catalase [70], which decomposes H_2_O_2_. Reactive oxygen species (ROS) are formed during sperm capacitation, which is important for the activation of CaMKII [12] and PLD [31]. Our group showed that treatment of bovine sperm with 50 µM H_2_O_2_ causes a significant increase in CaMKII phosphorylation/activation, a state that is completely reversed by 100 µM H_2_O_2_ [71]. In human sperm, the addition of SOD causes a decrease in the sAR [72]. In bovine sperm, hydrogen peroxide promotes capacitation, mimicking the role of bicarbonate in activating the soluble adenylate cyclase to activate the cAMP/PKA [73]. Also, ROS have been implicated in protein tyrosine phosphorylation, which mediates capacitation in several species [74]. Nevertheless, in boar sperm, ROS do not promote capacitation but stimulate the sAR [75].

In our recent study, we showed in human sperm that the activated tyrosine kinase FER enhances actin polymerization and protects sperm from the sAR [41]. Activated FER acts on several levels; it inhibits PP1 and regulates Ca^2+^ influx via CatSper, leading to CaMKII activation and actin polymerization. Simultaneously, FER also activates cortactin, leading to Arp2/3 activation and F-actin formation. In addition, FER regulates mitochondrial respiration via complex I of the electron transport chain [76] and restrains ROS production, thereby preventing CaMKII inhibition by high levels of ROS.

The knockout of several genes, including β-Defensin, the Lipocalin family LCN8 [51] or Aldehyde-dehydrogenase ALDH4A1, a key enzyme in mitochondrial prolin metabolism [77], results in an increase in mouse sperm sAR levels. However, the upregulation of cytochrome C in pig sperm promotes the sAR, indicating that mitochondrial activity stimulates the sAR [78]. This observation supports our notion regarding the regulation of the mitochondrial electron transport chain complex I by FER, whereby its inhibition promotes the sAR [41]. The regulation of the mitochondrial electron transport chain controls the production of ROS and protects the sperm from the sAR. Thus, FER, as an important regulator of mitochondrial activity is responsible for providing ATP for various sperm functions, leading to proper fertilization [41].

Interestingly, to prevent spermatozoa from potential oxidative stress damage, and probably from the sAR, the fatty-acid composition of rodent sperm membranes is altered by increasing the percentage of peroxidation-resistant fatty acids under competitive conditions [79].

Paraoxonase 1 (PON1) is a high-density lipoprotein-associated enzyme that acts as an antioxidant [80]. We showed that PON1 protects human sperm from the sAR [81]. Endogenous semen PON1 activity is negatively associated with the sAR, suggesting that PON1 protects against the sAR by reducing ROS levels [81]. It was also shown that a reduction in PON1 levels in semen is associated with infertility [82].

## 5. Role of Energy Metabolism in the sAR

It is well known that sperm ATP is produced by glycolysis and mitochondrial respiration. The inhibition of either glycolysis or oxidative phosphorylation in bovine sperm does not affect capacitation or sAR levels; however, when both systems are inhibited, no capacitation occurs, and there is a significant increase in sAR levels [48]. Under such ATP starvation, the increase in the sAR is triggered by Ca^2+^ influx into the sperm via the CatSper cation channel. There is no change in PKA activity when glycolysis or mitochondrial respiration is inhibited, while a complete reduction in PKA activity was observed when both systems were inhibited [48]. Protein tyrosine phosphorylation (PTP), also known to increase during sperm capacitation, was partially reduced by the inhibition of one metabolic system and completely blocked when the two metabolic systems were inhibited [48]. These studies show that ATP, PKA and PTP are involved in the mechanisms protecting sperm from the sAR.

In pig sperm, the levels of the β-subunit of H^+^-ATPase, the ATP-producing enzyme in the mitochondria, isocitrate-dehydrogenase (IDH) and pyruvate-dehydrogenase are enhanced during capacitation, while the level of enolase, a critical enzyme in anaerobic glycolysis, is decreased [78]. IDH is the main regulatory enzyme of the Krebs cycle, and its increase during capacitation indicates the involvement of the malate–aspartate shuttle required to maintain the levels of reduced NADP necessary for capacitation. Thus, mitochondrial and glycolytic activities are involved in the mechanism of sperm capacitation and protect sperm from the sAR.

## 6. Role of Zn^2+^ in the sAR

Zinc ions play an important role in the male reproductive system [83,84,85]. Zn^2+^ has antibacterial activity and can kill both Gram-positive and Gram-negative bacteria [86,87]. It was shown that Zn^2+^-deficient nutrition causes male infertility [88]. Zinc ions are secreted to the semen mainly from the prostate [89]. The addition of Zn^2+^ to semen extender before freezing sperm reduces ROS levels and increases the yield of fertilization after sperm thawing [90], but excess Zn^2+^ can increase ROS levels, resulting in the sAR [91]. It has been shown that capacitation-induced Zn^2+^ efflux allows sperm release from oviductal glycan by activating Zn-containing enzymes, such as metalloproteinase2, involved in sperm penetration of the ZP [92].

Extracellular Zn^2+^ interacts with the sperm Zn^2+^-sensing receptor (ZnR) also named GPR39 [93], which is localized in the sperm tail and the acrosome [94,95,96], suggesting its possible involvement in motility and the AR. We showed that Zn^2+^ mediates the GPR39-dependent bovine sperm AR [96] and human sperm hyper-activated motility/capacitation [94] (see Figure 2). Thus, the positive effect of Zn^2+^ on sperm capacitation protects sperm from the sAR [12].

GPR39 belongs to the G-protein-coupled receptor (GPCR) family, which activates trans-membrane adenylyl cyclase (tmAC), and the resulting cAMP activates PKA, a key enzyme in sperm capacitation. We demonstrated the involvement of two GPCRs, angiotensin-II-receptor and lysophosphatidic-acid receptor, in bovine sperm capacitation [97]. It has been suggested that Zn^2+^ can stimulate the activity of tmACas well as sAC [94] (see Figure 2). A relatively low concentration of Zn^2+^ (5 µM) causes an increase of about 40% in intracellular cAMP levels [96], but higher concentrations (20–30 µM) have a lesser effect [94]. These effects of different Zn^2+^ concentrations on intracellular cAMP levels correlate closely with the relatively higher stimulation of hyper-activated/capacitation at low concentrations of Zn^2+^ [94]. As described above, actin polymerization during sperm capacitation is essential for preventing the sAR. We found that the addition of 5 µM Zn^2+^ to bovine sperm increases the actin polymerization rate and decreases the sAR (unpublished). These results further support the significant effect of Zn^2+^ in protecting sperm from the sAR.

In Figure 2, we present a model summarizing the mechanisms that regulate human and bovine sperm capacitation mediated by Zn^2+^. The increase in intracellular cAMP by Zn^2+^ stimulates the Na^+^/H^+^ exchanger [98], which increases intracellular PH, leading to CatSper activation. Thus, the zinc ion stimulates hyper-activated motility/capacitation through the CatSper-dependent activation of the AC→cAMP→PKA→Src→EGFR and PLC cascade [94]. In bovine sperm capacitation, Zn^2+^ stimulates EGFR, which is mediated by the activation of tmAC→cAMP→PKA→Src [96]. In capacitated sperm, Zn^2+^ further stimulates the EGFR and the downstream effectors PI3K, PLC and PKC, resulting in the acrosome reaction [96] (see Figure 2). The hydrolysis of PIP_2_ by PLC generates IP_3,_ which activates IP_3_R localized in the outer acrosomal membrane and in the redundant nuclear envelope (RNE), resulting in Ca^2+^ release from the acrosome and RNE, promoting the development of hyper-activated motility/capacitation [99]. This cascade can be initiated by Zn^2+^-activated GPR39, leading to PLC activation.

## 7. Role of Protein Acetylation in the sAR

In a recent study, we showed that protein hyperacetylation protects bovine sperm from the sAR through an exchange protein directly activated by cAMP (EPAC) and via CaMKII-dependent and PKA-independent mechanisms [100]. Protein acetylation, including tubulin acetylation, is involved in sperm energy metabolism and motility [101]. Recently, several studies have described changes in the levels of acetylated proteins during human sperm capacitation [102,103]. Different protein acetylation profiles were observed in sperm during capacitation versus fertilization, suggesting that protein acetylation is involved in the fertilization process [103]. Changes in protein acetylation are also seen during axonemal microtubule construction [104,105], suggesting that poor sperm motility and male infertility may be associated with perturbed tubulin acetylation [105]. Moreover, hyperacetylation in non-capacitated mouse sperm induces capacitation-associated molecular events, including the activation of PKA and of the sperm-specific Ca^2+^-channel CatSper, hyperpolarization of the plasma membrane, hyperactivated motility and an increase in the AR [106]. Incubation of bovine sperm under non-capacitated conditions revealed a significant increase in the sAR that was reduced in the presence of deacetylase inhibitors, which caused protein hyperacetylation [100].

It was shown that EPAC mediates human and mouse AR [107,108]. We showed that the inhibition of PKA induces an EPAC-dependent AR [50]. Moreover, the induction of the AR by progesterone, angiotensin II, thapsigargin or Ca^2+^-ionophore is also mediated by EPAC, suggesting a physiological role of EPAC in the AR mechanism [100].

## 8. Additional Factors Regulating sAR

Several other factors were also shown to affect the sAR. These will be covered, in brief, below. Bacterial contamination: Bovine sperm incubated with the bacteria *Escherichia coli* (*E. coli*), *Staphylococcus aureus* (*S. aureus*) or *Pseudomonas aeruginosa* (*P. aeruginosa*) revealed a sperm–bacteria interaction; however, only *E. coli* and *P. aeruginosa* caused an increase in sperm sAR levels [109]. In addition, PKA and protein tyrosine phosphorylation activities were inhibited by the bacteria. Moreover, increasing intracellular cAMP, which also occurs during sperm capacitation, caused a significant reduction in the sAR induced by the bacteria [109]. Thus, the increase in the sAR by bacterial contamination in the semen or in the female reproductive tract could provide a possible explanation for infertility. It was further shown that disruption of in vivo β-defensins, an antimicrobial peptide, alters intracellular calcium levels, which leads to the sAR [110,111].

During induced AR and sAR in mouse sperm, IZUMO1 translocates from the acrosomal cap to the equatorial segment and further spreads over the whole sperm head. Moreover, protein tyrosine phosphorylation in the tail occurs at the beginning of the capacitation, and the progress of IZUMO1 relocation positively correlates with the level of acrosome instability, leading to the sAR [112]. In mammalian fertilization, IZUMO1 binds to its oocyte receptor counterpart, Juno, to facilitate recognition and fusion of the gametes.

sAR levels are elevated in CD46^−/−^ mice, indicating that CD46, which is localized in the inner acrosomal membrane, plays a role in sperm protection against the sAR [113,114,115]. Human membrane cofactor protein CD46 is a ubiquitously expressed protein known to protect cells from complement attack. The absence of CD46 protein expression is associated with acrosomal instability in mice expressing novel CD46 transcripts, resulting in a rapid sAR. This provides a strategy to increase the competitive sperm advantage for individuals, leading to faster fertilization in highly promiscuous genus.

In *Lcn8*^−/−^ male mice, the proportion of immobilized sperm was elevated in the cauda epididymis, and the sperm sAR frequency was increased [116]. Lcn8 is a member of the lipocalin family, in which *Lcn8*^−/−^ and *Lcn9*^−/−^ male mice showed normal spermatogenesis and fertility, while a decreased sperm quality was noticed in *Lcn8*^−/−^ male mice, including increased morphologically abnormal sperm, attenuated sperm motility and premature acrosome reaction of the sperm in cauda epididymis. These results indicated that Lcn8 deficiency causes epididymal sperm maturation defects in mice.

Sperm express epidermal growth factor receptor (EGFR), and the AR can be induced by EGF, indicating that PLCγ is required for the AR [97]. We also showed that α7 nicotinic acetylcholine receptor (α7nAChR) might be a sperm receptor for the interaction with the oocyte, activated by solubilized ZP to induce an EGFR-mediated AR. Isolated ZP or α7 agonists induced the AR in sperm from WT but not in α7-null mouse spermatozoa, and the induced AR was inhibited by α7 or EGFR antagonists. Moreover, the sAR in α7-null sperm was very low, indicating that the regulation of the AChR serves as a protective mechanism against the sAR [117]. This conclusion was further supported in a more recent study showing that AChR antagonists suppress the sAR induced by acetyl-choline or nicotine [118]. In human sperm, nicotine causes an increase in the sAR; thus, the occurrence of high levels of nicotine in the body and, specifically, in seminal fluid might affect fertilization capacity [119].

These findings that occur in the sAR are vital for understanding the fertilization process.

## 9. Conclusions

Spermatozoa contain several protective mechanisms that limit the sAR. In general, any defect in a mechanism leading to sperm capacitation might promote the sAR, and, accordingly, some of the processes that occur in capacitation are protective. Such mechanisms include PKA, PLD, CaMKII, tyrosine kinase activities and actin polymerization. Further elucidation of these mechanisms should enable us to optimize fertilization both for IVF in humans and for animal breeding.

## Figures and Tables

**Figure 1 ijms-24-17005-f001:**
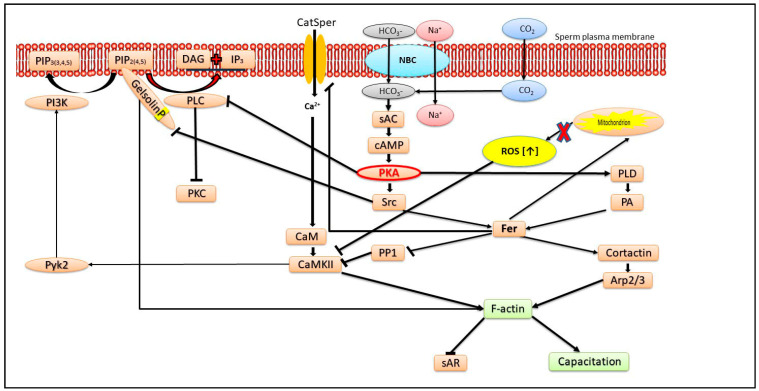
The mechanisms that protect sperm from sAR: All pathways that lead to F-actin formation and complete capacitation protect sperm from sAR. The main factor is protein-kinase A (PKA) activated by cAMP generated by HCO_3_-activated soluble adenylyl-cyclase (sAC). The initial entrance of HCO_3_^−^ is mediated by Na^+^/H^+^ cotransporter belonging to the SLC4 family or by the conversion of CO_2_ to HCO_3_^−^ by carbonic-anhydrase. PKA inhibits phospholipase C (PLC), preventing phosphatidyl-inositol-4,5-bisphosphate (PIP_2_) hydrolysis and promoting the interaction between gelsolin and PIP_2_, leading to gelsolin tyrosine phosphorylation by Src and maintaining its inactive state allowing F-actin formation. The activation of Src and phospholipase D1 (PLD) by PKA also causes Fer activation, which promotes F-actin formation by two pathways: Fer activates cortactin, leading to Arp2/3 activation, and also inhibits the protein phosphatase 1 (PP1), causing calmodulin-kinase II (CaMKII) activation, resulting in F-actin formation. Fer also prevent production of too much ROS by the mitochondria and regulates Ca^2+^ transport via CatSper, leading to CaMKII activation and preventing relatively high Ca^2+^ influx, which causes sAR. Active CaMKII activates the tyrosine kinase Pyk2, which phosphorylates phosphtidyl-inositol-3-phosphate (PI3K), promoting PIP_2_ formation and increasing the opportunity of gelsolin to interact with PIP_2_, which causes gelsolin inhibition as described above. In the flagellum, Fer also regulates complex I of the mitochondrial electron transport chain, thereby restraining ROS production and preventing CaMKII inhibition.

**Figure 2 ijms-24-17005-f002:**
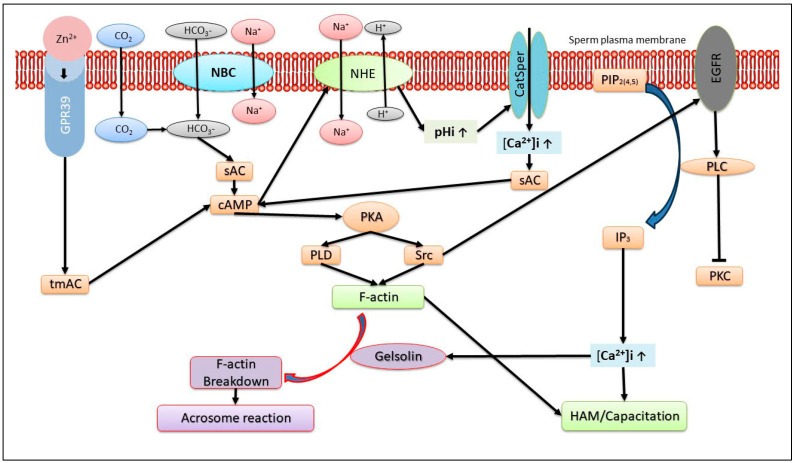
The mechanism of action of Zn^2+^ in sperm capacitation and the acrosome reaction. The binding of Zn^2+^ to the sperm receptor GPR39 activates trans membrane adenylyl-cyclase (tmAC) to produce cAMP, which activates the Na^+^/H^+^ exchanger (NHE), resulting in intracellular alkalization and CatSper activation. The elevation of intracellular Ca^2+^ concentration together with HCO_3_^−^ activates the soluble adenylyl-cyclase (sAC), which further enhances the level of cAMP, leading to protein-kinase A (PKA) activation. PKA activates the tyrosine-kinase Src, followed by epidermal growth factor receptor (EGFR) activation, and that of its downstream effector, phospholipase Cγ (PLC), which catalyzes the hydrolysis of phosphatidyl-inositol-4,5-bisphosphate (PIP_2_) to produce inositol-tri-phosphate (IP_3_) and diacylglycerol. IP_3_ activates the IP_3_ receptor of the outer acrosomal membrane to release Ca^2+^ from the acrosome to the cytosol, promoting the development of hyper-activated motility and capacitation. PKA also activates phospholipase D1 (PLD), which promotes F-actin formation and capacitation. At the end of capacitation and prior to the acrosome reaction, there is an increase in intracellular Ca^2+^ concentration to the µM range, which further activates PLC to hydrolyze PIP_2_, inducing the release of PIP_2_-bound gelsolin, which is activated by Ca^2+^, and breakdown of F-actin, enabling the acrosome reaction.

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
