# Peer review of "Mechanisms That Protect Mammalian Sperm from the Spontaneous Acrosome Reaction"

_ijms, 2023, doi:10.3390/ijms242317005_

Round 1

Reviewer 1 Report

Comments and Suggestions for Authors

Mammalian spermatozoa require a series of biochemical reactions to occur in the female reproductive tract, collectively known as capacitation, to be able to fertilize an egg. During capacitation, sperm become capable of undergoing the acrosome reaction and interacting with the egg's zona-pellucida. However, if the acrosome reaction happens prematurely (sAR) long before reaching the egg, it can hinder fertilization. A key aspect of capacitation involves actin polymerization, thereafter, the resulting F-actin must disperse for the acrosome reaction to happen. The changes involved in sperm capacitation, including actin polymerization, serve to protect sperm from the spontaneous acrosome reaction (sAR). The present review outlines the mechanisms that regulate sperm capacitation to safeguard against premature sAR, ensuring successful fertilization.

The subject of this review is important and relevant for the field of reproduction. It presents a summary, mostly of the work carried out by the Breitbart group, regarding the capacitation processes that are necessary for this process but in addition prevent sAR. Before being published, the review requires significant editing and improvements in how it is written. More importantly, it needs an update on the evidence of where the acrosome reaction occurs, as this is key to the processes that safeguard sAR. It would also be more helpful to the field if it included final sections on conclusions and perspectives that would include key unsolved problems.

In what follows I include suggestions, indicate some errors, and present some questions either in yellow or in blue in the pdf which I attach.

1)     The statements on lines 26-27 and 52 are debatable, certainly in mouse sperm. The authors need to update this information as AR occurs before reaching the egg (see Hirohashis' work etc).

2)     Define E2 on line 39.

3)     The sentence in lines 73-74 needs to be rewritten.

4)     In Fig. 1 what activates HCO3- uptake, how is it transported and regulated? Please discuss Grhan et al. Nature 2023. Please explain the evidence that Fer regulates CatSper.

5)     Please rewrite the sentence on lines 152-153.

6)     The statement on line 197 is wrong, please correct.

7)     Please explain how Zn2+ stimulates Na+/H+ exchange, and which exchanger is involved?

Comments on the Quality of English Language

Before being published, the review requires significant editing and improvements in how it is written. More importantly, it needs an update on the evidence of where the acrosome reaction occurs, as this is key to the processes that safeguard sAR. It would also be more helpful to the field if it included final sections on conclusions and perspectives that would include key unsolved problems.

Author Response

Reviewer # 1: See marked lines in yellow

Conclusions were added (see lines 338-344)

1) Lines 26-27 and 52: Information regarding the site of AR was added (see lines 28-30 ).

2) E2 was defined (Line 52)

3) Lines 73-74 were corrected (see lines 87-91)

4) This point was discussed in the legend to Fig. 1 (see lines 350-352)

5) Lines 152-3 were corrected (see lines 169-171)

6) Line 197: See correction (lines 159-160)

7) See explanation in lines 366-369.

Reviewer 2 Report

Comments and Suggestions for Authors

This review discusses the known mechanisms of spontaneous acrosomal exocytosis in various mammalian species. The review has good informational value, is mostly well-written, and easy to follow and comprehend. This reviewer suggests having English edited by a native speaker. This reviewer does not see a major concern with the MS and offers suggestions that may be implemented by the authors at their discretion.

Title: Please, include what spermatozoa it relates to, as in species of the group of species, i.e. bovine spermatozoa, mammalian spermatozoa, etc.

L10: please use spermatozoon, spermatozoa, and sperm consistently throughout the MS. In general; spermatozoon is a noun in singular form, spermatozoa is a noun in plural form, and sperm is an adjective.

L21: and elsewhere: please refrain from using the term egg as it indicates a fully matured female gamete. At the gamete encounter, the female gamete is in the MII arrested stage. I recommend using an oocyte instead of an egg.

L27: the egg/oocyte surrounding is called cumulus oophorus, please you the established nomenclature

L30: please, add “at least in this species” as the mechanism can differ in other species. This reviewer also encourages the author to indicate which species the discussed piece of information relates to in the rest of the MS. This will also demonstrate that the authors are aware that the mechanisms can differ across different species.

L35: Please, briefly explain spontaneous acrosomal exocytosis, and how it differs from physiological AR.

L38: Please, spell out ROS at its first occurrence.

L41-2: How do they influence spermatogenesis? What consequences will it have for fertilization? be more specific, please.

L42: Please, state the relevance of 2AG for male reproduction.

L47, 292: Please, specify what inducer of AR was used.

L55: What do the authors mean by resting spermatozoa? As in the dormant sperm that are attached to the oviductal epithelium reservoir? The authors can also discuss how intact acrosomes are important for the oviductal sperm reservoir as the sperm receptors that recognize oviductal epithelium ligands are in the periacrosomal plasma membrane.

L55-6: The last two sentences are confusing. The first sentence indicates that the authors are not sure whether those mechanisms exist, while the second sentence implies that there are. I suggest rewriting the two sentences as the goal of this review is to describe the known mechanisms that protect spermatozoa from sAR.

L61 and elsewhere: protein tyrosine kinase; specifically for L61: please, be more specific about which protein tyrosine kinase it is.

L64: Please, change to “capacitating conditions”

L66: This reviewer does not see PKC in Fig. 1. Please, amend.

L66-68: This sentence makes little sense to this reviewer since the previous sentence states that PLC is inactivated by PKA. Please, elaborate under what conditions this is valid, or reformulate.

L71: Consider changing “functions” to outcomes or results or effects

L76: Please add: Stimulation of capacitation, measured by chlortetracycline assay, and inhibition…

L80: during incubation in synthetic culture media? As in during in vitro capacitation? Please, change it if accurate.

L80-4: The authors need to consider that sperm in [37] we normally capacitating (as shown by CTC assay), but were unable to undergo AR, while in [30] the authors were inhibiting PKA-the central molecule of capacitation (see Fig. 1 of this paper). You are comparing capacitated with most likely non-capacitated spermatozoa here. Please, make the distinction in the text.

L81: This reviewer suggests adding the outer acrosomal membrane in the figure so that the process compartmentalization is obvious as well.

L91: please, change to protein phosphatase 1

L96: Consider starting the sentence with. In the flagellum, Fer also regulates.... since the figure presumably depicts the sperm head.

L137: There are multiple dismutases in spermatozoa, such as SOD1 and SOD2, please amend.

L187: Please, consider changing “support” to “trigger” as sAR is not desirable.

L187,9: Please, state the molarities of relative low and high Ca2+ concentrations.

L189: decreases the protein tyrosine phosphorylation levels

192: please, specify where Ca2+ from acrosome is released to.

L193: influx means from the external environment to the internal cellular space, so in this scenario from the external environment into intermembrane space or to the acrosome? How this reviewer reads the sentence is that inhibition of the Ca2+ ATPase results in Ca2+ efflux from the acrosome into the extracellular space, followed by Ca2+ influx from the extracellular space to the acrosome which triggers sAR. Please, reformulate the sentence for better clarity.

L196: Elaborate how NMDA-type glutamate receptor mediates sAR.

L197: inhibition of the sperm-specific potassium channel

L204 and elsewhere: please distinguish between spontaneous and physiological/induced acrosomal exocytosis so that the reader does not need to guess.

L212 and elsewhere: if the data was presented at a scientific proceeding, please provide where it was, for instance: the results were presented at the 56th Annual Meeting of the Society for Study of Reproduction, 11-14 July 2023 in Ottawa, CAN.

 L223: Zinc has a biphasic effect on sperm capacitation depending on the concentration, please see doi: 10.3390/ijms21062121   

 Fig 2: What species is this model valid for? Same as for Fig. 1, the authors can add the outer acrosomal membrane to the figure as well. The authors can add proton channel hv1/HVCN1 to the figure that helps with alkalinization if relevant.

L256: How does a Ca2+ release from the acrosome promote flagellar hyperactivation since these are two different sperm compartments? Please, elaborate.

L256: PLD1 vs PLD in Fig. 2. Please, make it consistent.

L286: What do the authors mean by the reverse of sAR? To this reviewer's knowledge, acrosomal exocytosis is an irreversible process. Please, reformulate.

L290: premature activation of what exactly? As in premature capacitation? Please, reformulate.

L295-6: The events that occur during sAR are vital...

L298: please change “protects sperm from sAR” to “is important for sperm protection against sAR”

L298-9: This reviewer does not understand what the authors are trying to say in the second part of the sentence. ... in mice expressing novel CD46 transcripts... how they can do so if they are CD46 knockouts plus spermatozoa are transcriptionally and translationally silent. Please, reword.

L302-3: This reviewer acknowledges what the authors are saying, but is it physiologically relevant, though?

L380: Which ZP glycoprotein? Or rather solubilized zona pellucida as a whole? Please, specify in the text.

L309: Please, specify the species.

L315: Please, add a concluding paragraph to summarize the review.

Comments on the Quality of English Language

Have the MS edited by a native English speaker.

Author Response

Reviewer # 2: See marked lines in green

English was edited by English speaker.

Title: the word Mammalian is now included.

L10 was corrected along the manuscript.

L21: the word “egg” was changed to “oocyte”.

L27: Corrected (see line 72)

L30: The names of the species were added.

L35: AR and sAR are now defined (see lines 43-7)

L38: ROS was spelt out (lines 50- 51)

L41 : Effect of HFD on spermatogenesis, (see Mu Y. et al).

L42 : The relevance of 2AG is now described (lines 58-60)

L47: The AR inducer is A23187 (see lines 65- 66)

L55 : This point was corrected (line 76-8)

L55-6 : See correction (line 76-8).

L 61 : The names of 3 tyrosine kinases were presented (line 84)

L64 : corrected to capacitating conditions.(line 86)

L66 : PKC was added to Fig. 1.

L66-68 : In this sentence we explain the role of PLC in PKC activation. It is better described now (lines 88-92)

L71 : corrected (line 96)

L76: Deleted

L80: The medium used is Earl’s salt solution contained 10% maternal serum (EBSS).

L80-4 : See explanation (lines 106-7)

L81: Adding the outer acrosomal membrane to the figure would make it very complicated to understand, we prefer not to do it.

L91 : corrected (line 116)

L96 : Corrected as suggested (line 370-2)

L137 : We do not know for sure if it SOD1 or SOD2.

L187: Was changed as suggested (line 147)

L187,9: The Ca concentrations are now described (lines 148-9)

L189: corrected (line 150)

L192 : Was explained (lines 153-4)

L193: This point is now explained better (lines 153-5)

L196 : See explanation (line 158-9)

L197:Corrected (line 159)

L204: Done.(see line 169)

L212 : These data were not presented in a meeting.

L223: Information on [Zn] was added(see lines 246-8 and 260-2).

Fig.2 : The model is valid for human and bovine sperm.

L256: Ca can be transported from the head to the tail.

L256: corrected. (line 376)

L286: We aim to say “reduced” , corrected (see line 309)

L290 : Corrected (line 314-5)

L295-6: corrected (lines 317-8)

L298 : Was changed as suggested (lines 319-320)

L298-9: This point is now corrected (see lines 322-3 )

L302-3: Prolactin receptor was detected in sperm .

L308: Corrected (line 329)

L309: The species was specified :line 330)

L315 : Concluding paragraph was added at the end of the ms.(lines 338-344)

The manuscript was edited by English expert.

Reviewer 3 Report

Comments and Suggestions for Authors

The present review is very interesting in addressing the mechanisms that give rise to or prevent the spontaneous acrosome reaction, however, in my opinion the manuscript is not well organized or structured.

The manuscript only consists of two sections: 1) Introduction and 2) Role of actin polymerization and various enzymes.

The introduction should be rewritten, too many very specific data are given without connection, and it does not focus on the topic to be discussed.

The rest of the manuscript is included under the heading “2. Role of actin polymerization and various enzymes” and all the other subsections go within the latter. In some of these subsections, some paragraphs begin with headings, such as "role of ezrin in sAR" (line 124), "effect of NOS" (line 167) or “H+-transport mechanisms” (line 208), followed by a colon and a short explanation. In my opinion, these titles are unnecessary, and the paragraphs should be included as normal paragraphs in the context of the corresponding section, or if they have their own entity, be part of a separate section.

Unquestionably, all the information should be organized in another way. Perhaps, (it is only a suggestion) starting with the importance of actin polymerization, in a more general way, which could even go in the introduction since it is practically the main event of the entire manuscript. It could include the main direct effectors and inhibitors (gelsolin..). And the pathways, ions, enzymes, ROS... involved in its regulation, going in separate sections with their own entity.

And perhaps the effectors not related to actin polymerization (or the mechanism of action is unknown) could go in another section... These are just suggestions but the manuscript should be better organized into sections and the ideas within each section should be more connected and avoid repetitions.

And, finally, a conclusion or final summary would be necessary, otherwise the manuscript remains unfinished.

Regarding bibliographic references, only 20% are from the last five years (2018-2022+2023). Perhaps more current bibliography should be included since the manuscript is a review.

There are some sentences that are not very connected to the rest of the paragraph. For example, paragraph from line 196 to line 199: “ NMDA-type glutamate receptor mediates sAR in newt sperm [58]. Furthermore, in human sperm, inhibition of the current of the sperm-specific potassium channel, KSPER, which is predominant for Ca2+ influx in sperm, decreased the level of sAR [72]. Extracellular ATP stimulates Ca2+ influx and sAR via operation of P2y purinergic receptor and PKC activation [73].” In my opinion it is disjointed it's hard to follow a common thread.

Also the sentence in line 212 “Sperm cytosol alkalization lead to CatSper activation [84] and an increase in sAR(our unpublished data)”  is disconnected from the rest of the text.

Paragraph from line 79 to 83 is not understood if it is not better explained: aryl hydrocarbon receptor -KO spermatozoa? CDC42 abundance? What is the relationship with the topic discussed?

The last subsection (Various effects on sAR) should be rewritten in a different way, not as a list of proteins with specific effects.

Other specific issues:

Line 13: “One of the main processes in capacitation involves the process of actin polymerization and the produced F-actin should be dispersed prior to the acrosome reaction”. Please, avoid the repetition of the word “process”.

 Line 14: “All biochemical reactions that occur during sperm capacitation including actin polymerization, protect sperm from sAR”. I don´t agree with this statement. All biochemical reactions protect sperm from sAR?

Throughout the entire manuscript there are words that are written together, without separation between them, for example, in lines 26, 27, 38....

What is the meaning of “sperm from polyspermic men”? (line 47)

What is the meaning of “resting sperm”? (line 55)

In heading “Role of sperm metabolism in sAR” (line 169), I would recommend including the word “energy”, that is, “Role of energy metabolism in sAR”, or better, “Relationship of energy metabolism with sAR”.

“Role of EPAC” could be included together with “Role of protein acetylation in sAR” (if the latter is maintained as a section).

“Role of Ezrin in sAR” (line 124) is a subsection?

Are the models in figures 1 and 2 related to human sperm (as mentioned in line 236) or is general for all species?

In the legend of the figures, all acronyms should be explained with the complete words and the acronyms in parentheses, not just some of them. The same throughout the entire manuscript, at least the first time they are mentioned (E2, MUPP1, PI4k, FER..).

Line 286: “…caused significant reverse of sAR induced by the bacteria sAR”. What is the meaning of this sentence?

Lines 294- 295: “and the progress of IZUMO1 relocation which is positively correlate with the level of promiscuity and the acrosome instability in promiscuous species”. What is the meaning of this sentence?

Comments on the Quality of English Language

It is quite well written.

Author Response

Reviewer # 3: See marked lines in light blue.

There are more sections in the ms.

The introduction was corrected

The suggested subsections were deleted.

Final conclusions were added at the end of the manuscript (see lines 338-344)

Seven recent references were added

The paragraph (lines 196-9) was rewritten (lines 158-160)

Line 212 is now better explained (see lines 179-181)

Lines 79-83 : in this paragraph we describe data which contradict our concept in bovine sperm.

The subsection :”Various effects on sAR “ was rewritten (line 302)

Line 13 :Corrected as suggested (lines 14-16)

Line 14 :The word “all” was changed to “several” (line 15)

Words written together were separated along the manuscript.

Line 47 : mistake : should be polyzoospermic (see line 65)

Line 55 : The sentence was deleted .

Line 169: The word “energy “ is now included (line 223)

The role of EPAC is now included in "Role of protein acetylation"(lines298-301).

Line 124 : Ezrin role is now described in lines 137-145.

Fig. 1 is general and Fig. 2 describe the situation in human and bovine sperm.

All acronyms in the Fig legends were explained.

Line 286 :the word “reverse “ was changed to “reduction” (line 309)

Lines 294-5: The sentence was corrected (see lines 314-5).

Round 2

Reviewer 3 Report

Comments and Suggestions for Authors

The manuscript has been substantially improved. The authors have listened to almost all my requests.

I only have a few comments:

1)     I still don't understand the sentence “Moreover, aryl hydrocarbon receptor-KO spermatozoa were frequently capacitated, but showed reduced spontaneous and progesterone-induced acrosome reaction, which is related to low CDC42 abundance and very limited actin polymerization during capacitation [36]” and its relationship with the rest of the paragraph (now in the end of page 6 in the revised manuscript).

2)     The title in section 6 should be: “Role of Zn2++ in sAR” to be consistent with the rest of the titles of the manuscript.

3)     In section 8 “Additional factors regulating sAR”: In 3rd paragraph “This provides a strategy to increase the competitive sperm advantage for individuals, leading to faster fertilization in this highly promiscuous genus” Are you referring to mice? Then it is better to use the term “species” or to change the phrase to “… leading to faster fertilization in highly promiscuous genus”.

4)     In the same section: IZUMO1, CD46 or Lcn8 should be briefly explained/described before to relating them to the sAR.

Author Response

1) This sentence shows that there is reduced sAR in spite of low actin polymerization (see lines 101-3).

2) Corrected (Line 240)

3) Corrected (Line 328)

4) Brief explanations were added: IZUMO1- lines 316-17; CD46- lines 319-21 and Lcn8-lines 326-31.